# Critical COVID-19, *Victivallaceae* abundance, and celiac disease: A mediation Mendelian randomization study

Yuxin Zou[1], Manyi Pan[1], Tianyu Zhou[1], Lifeng Yan[1], Yuntian Chen[1], Junjie Yun[1], Zhihua Wang[1], Huaqi Guo [1]*, Kai Zhang[2]*, Weining Xiong[1]*

1 Department of Respiratory and Critical Care Medicine, Shanghai Ninth People's Hospital, Shanghai Jiao Tong University School of Medicine, Shanghai, China, 2 Department of Public Health, Shanghai Jiao Tong University School of Medicine, Shanghai, China

* xiongdoctor@qq.com (WX); zhangk19@sjtu.edu.cn (KZ); guohuaqi814102@163.com (HG)

**Data Availability Statement:** The datasets underpinning the analyses presented in this paper are openly available in dedicated repositories.

## Abstract

Celiac disease exhibits a higher prevalence among patients with coronavirus disease 2019. However, the potential influence of COVID-19 on celiac disease remains uncertain. Considering the significant association between gut microbiota alterations, COVID-19 and celiac disease, the two-step Mendelian randomization method was employed to investigate the genetic causality between COVID-19 and celiac disease, with gut microbiota as the potential mediators. We employed the genome-wide association study to select genetic instrumental variables associated with the exposure. Subsequently, these variables were utilized to evaluate the impact of COVID-19 on the risk of celiac disease and its potential influence on gut microbiota. Employing a two-step Mendelian randomization approach enabled the examination of potential causal relationships, encompassing: 1) the effects of COVID-19 infection, hospitalized COVID-19 and critical COVID-19 on the risk of celiac disease; 2) the influence of gut microbiota on celiac disease; and 3) the mediating impact of the gut microbiota between COVID-19 and the risk of celiac disease. Our findings revealed a significant association between critical COVID-19 and an elevated risk of celiac disease (inverse variance weighted [IVW]: $P = 0.035$). Furthermore, we observed an inverse correlation between critical COVID-19 and the abundance of *Victivallaceae* (IVW: $P = 0.045$). Notably, an increased *Victivallaceae* abundance exhibits a protective effect against the risk of celiac disease (IVW: $P = 0.016$). In conclusion, our analysis provides genetic evidence supporting the causal connection between critical COVID-19 and lower *Victivallaceae* abundance, thereby increasing the risk of celiac disease.

## Introduction

The influence of coronavirus disease 2019 (COVID-19) extends beyond respiratory symptoms, with emerging evidence linking it to various system [1], encompassing cardiovascular [2, 3], endocrine [4, 5], autoimmune [6–8], and gastrointestinal system [9]. Research on the systemic

Specifically, the data related to COVID-19 are accessible through the COVID-19 Host Genetics Initiative at (https://www.covid19hg.org). Data pertaining to gut microbiota are available at the MiBioGen initiative portal (http://www.mibiogen.org). The datasets for celiac disease genetics can be found in the GWAS Catalog at (https://gwas.mrcieu.ac.uk/datasets/ieu-a-276). Additionally, the MR analysis code employed in this study is available at https://mrcieu.github.io/TwoSampleMR/articles/index.html. These resources support the findings and reproducibility of the work presented in this paper and its Supporting Information files.

**Funding:** The author(s) received no specific funding for this work.

**Competing interests:** The authors declare that they have no known competing financial interests or personal relationships that could have appeared to influence the work reported in this paper.

effects of COVID-19 has expanded the focus to investigate its impact on diverse physiological systems beyond the respiratory tract, notably targeting the gastrointestinal system [10, 11]. Notably, celiac disease, a chronic autoimmune gastrointestinal disorder, has garnered significant attention due to their higher prevalence among COVID-19 patients [12, 13].

Celiac disease commonly occurs in individuals genetically predisposed to gluten consumption [14]. Nonetheless, genetic factors and a gluten-rich diet are not exclusive contributors in celiac disease development. Studies have highlighted a connection between gut dysbiosis, an imbalance in gut microbiota, and the onset of celiac disease [15, 16].

Recently studies have highlighted significant alterations in the gut microbiota among individuals affected by COVID-19 [17, 18]. Considering the association among gut microbiota alterations, COVID-19 and celiac disease, the gut microbiota may emerge as potential mediators underlying the impact of COVID-19 on celiac disease. COVID-19 may contribute to the initiation or exacerbation of celiac disease in susceptible populations, potentially via disturbances in gut microbiota.

Understanding the potential influence of COVID-19 on the development or exacerbation of celiac disease, particularly through mechanisms involving gut dysbiosis, has significant implications. Further investigations into these interactions are crucial to unravel the underlying mechanisms and potentially devise targeted interventions to mitigate the impact of COVID-19 on individuals susceptible to or affected by celiac disease.

Most classical observational studies often face challenges regarding reverse causation and confounding factors. Mendelian randomization (MR) [19], a well-established method for assessing causation between exposure and outcomes, offers numerous advantages when exploring causality [20]. Single-nucleotide polymorphisms (SNPs) identified in genome-wide association studies (GWAS), which are significantly linked to an exposure and independent of several confounders, were utilized as instrumental variables (IVs). We conducted a two-step MR analysis to ascertain the genetic causal association between COVID-19 and celiac disease, and investigated the potential intermediary role of gut microbiota in this causality [21].

## Materials and methods

MR studies need to satisfy three essential criteria for the proper utilization of valid genetic variants, particularly SNPs, as IVs: the relevance assumption, which requires a robust association between genetic variants and the targeted exposure; the independence assumption, indicating that genetic variants should be independent of any confounders that may affect the relationship between the exposure and outcome; and the exclusion restriction assumption, affirming that SNPs influence the outcome solely through the exposure pathway [22].

### Study design

Fig 1 illustrates the study design of two-step MR analysis. The first step involved: 1) selecting genetic variants related to COVID-19 as IVs, categorizing COVID-19 severity into three groups: COVID-19 infection, hospitalized COVID-19 and critical COVID-19; 2) identifying celiac disease as the outcome; and 3) performing Mendelian randomization analysis to estimate the causal effects of COVID-19 infection, hospitalized COVID-19 and critical COVID-19 on celiac disease. The second step included:1) identifying a significant causal effect of critical COVID-19 on celiac disease following the initial analysis, hence selecting critical COVID-19 as the exposure; 2) choosing gut microbiota as potential mediators; 3) selecting genetic variants related to gut microbiota; and 4) performing a MR analysis to estimate the causal effect of critical COVID-19 on celiac disease, while exploring the potential mediating role of gut microbiota.

## Two-step Mendelian randomization analysis

**Fig 1. Study overview.** The two-step MR analysis consists of two steps. In the first step, SNPs associated with COVID-19 infection, hospitalized COVID-19 and critical COVID-19 are chosen to estimate their causal effects on celiac disease. These selected SNPs should be independent of unknown confounders and exclusively affect the outcome through exposure. In the second step, a separate set of SNPs related to gut microbiota is identified. MR analysis is then employed to estimate the causal relationships between critical COVID-19, gut microbiota and celiac disease.

### Data source

In our primary analysis, two-step MR analysis was applied to study the association of COVID-19 and gut microbiota on the risk of celiac disease. The COVID-19 Host Genetics Initiative (https://www.covid19hg.org/, Release 7, European ancestry cohorts) provided GWAS summary statistics for severe respiratory symptom (A2: critically ill COVID vs. population), hospitalization (B2: hospitalized COVID vs. population) and COVID-19 infection (C2: COVID vs. population). The GWAS dataset comprised 13,769 cases vs. 1,072,442 controls (A2), 32,519 cases vs. 2,062,805 controls (B2), and 122,616 cases vs. 2,475,240 controls (C2).

Summary statistics for gut microbial taxa were obtained from an extensive GWAS analysis involving 18,340 participants with diverse ancestries and ages across 24 cohorts [23]. The GWAS analysis employed a standardized 16S rRNA processing pipeline for microbiome data, including rarefying samples to 10,000 reads, binning reads to a reference database via the RDP classifier, and limiting taxonomic resolution to the genus level. To meet the study-wide standards for mbQTL mapping, taxa needed to be present in at least 10% of the samples across three cohorts and require an effective sample size of at least 3,000 samples. Consequently, 211 taxa (131 genera, 35 families, 20 orders, 16 classes, and 9 phyla) met the inclusion criteria for mbQTL analysis. Following adjustments for age, sex, technical covariates, and genetic principal components, Spearman correlation analyses were conducted to pinpoint genetic loci influencing bacterial taxa abundance within the microbiome. The complete statistics of the association study are available at the www.mibiogen.org website.

Our study utilized the comprehensive GWAS on celiac disease, analyzing 523,399 SNPs within a cohort comprising 4,533 celiac disease patients and 10,750 healthy individuals, all of European ancestry [24]. Rigorous quality control measures were systematically applied to exclude samples with low call rates, genotype-inferred gender, ethnic outliers and duplicates. Principal component analysis was utilized to evaluate sample quality and probe-specific variance excluding lower quality samples. These steps are fundamental in ensuring data reliability and integrity of data in the quality control process of celiac disease GWAS study. Access to a summary of this dataset is freely available at https://gwas.mrcieu.ac.uk/datasets/ieu-a-276. To prevent pleiotropic effects of cross-lineage cases [25], we exclusively utilized data from study participants of European ancestry. Further comprehensive details regarding the datasets are listed in S1 Table in S2 File.

## Selection of instrumental variables

The identification of independent genetic IVs followed stringent procedures. Firstly, SNPs significantly associated with exposure were required to meet the genome-wide significance threshold of $P < 5E-08$. Secondly, to prevent linkage disequilibrium (LD) clumping among IVs, only independent SNPs with $r^2 < 0.001$ and kb = 10,000 kb were selected. Thirdly, SNPs correlated with the outcome and palindromic SNPs were excluded. Our investigation utilized the phenoscanner database (http://www.phenoscanner.medschl.cam.ac.uk/) to exclude SNPs associated with potential confounders. In line with the MR analysis hypothesis, the selected SNPs should exhibit strong correlation with the exposure. The *F*-statistic, commonly used to assess the strength of IVs associated with exposure, indicates a weak correlation between IVs and exposure when its value is below 10 [26]. Consequently, we exclusively selected SNPs with *F*-statistics greater than 50 to avoid weak instrumental bias.

## Sensitivity analysis and MR analysis

The presence of horizontal pleiotropy effect, where genetic variation influences outcomes through other pathways rather than specific exposures, may lead to inaccurate findings. Employing the MR Egger intercept and MR Pleiotropy Residual Sum and Outlier (MR PRESSO) methods helped identify potential horizontal pleiotropy [27]. The Cochran's *Q* test of the IVW and MR Egger methods was used to assess heterogeneity [22, 28]. A significance level ($P > 0.05$) indicated the absence of significant pleiotropy or heterogeneity among the genetic IVs.

The causal relationship between exposures and outcomes was determined using five MR methods: MR egger, weighted median, IVW, simple mode and weighted mode methods, prioritizing IVW as the primary method and considering exposures and outcomes as causally related at a significance level of $P < 0.05$. To further assess the robustness of the identified causal associations, Steiger test and leave-one-out analyses were executed. The Steiger test aimed to identify the causal directionality by comparing the variances ($R^2$) of exposures and outcomes [29]. Leave-one-out analyses helped detect and exclude any potential outliers that could bias overall causal estimations. All statistical and sensitivity analyses were performed using the TwoSampleMR package (v 0.5.5) within the R software (v 4.2.1).

To exploring the mediating effect of gut microbiota in the relationship between COVID-19 and celiac disease, a two-step MR analysis was conducted. The mediating effect of gut microbiota on this association were calculated using β values obtained from IVW methods. Here, $\beta_1$ represents the causal effect of COVID-19 on celiac disease, $\beta_2$ signifies the causal effect of COVID-19 on gut microbiota, and $\beta_3$ indicates the causal effect of gut microbiota on celiac disease. Mediator effects were calculated as $\beta_2 \times \beta_3$. The intermediary percentage ($\beta_0$) was

derived from $(\beta_2 \times \beta_3)/\beta$, representing the proportion of mediator effects to the total effect.

$$\beta_0 = \frac{\beta_2 \times \beta_3}{\beta_1}$$

**Ethics statement.** The GWAS data used in this article are anonymous and open access, available on the web, so it was not essential to obtain the ethical approval by the institutional review board of Shanghai Ninth People's Hospital, Shanghai Jiao Tong University School of Medicine.

## Results

### Causal effect of COVID-19 on celiac disease

After implementing a series of rigorous quality control measures, we obtained strongly correlated independent SNPs for exposure ($P < 5E-08$, $r^2 < 0.001$). Specifically, 4 SNPs were associated with COVID-19 infection, 6 SNPs were associated with hospitalized COVID-19, and 7 SNPs were associated with critical COVID-19 (Fig 2). The IVs exhibited *F*-statistics ranging from 53.07 to 183.85, all surpassing 10, indicating robust instrumental strength and no weak instrumental bias (listed in S2 Table in S2 File). The Cochran's *Q* test of IVW and the MR Egger methods revealed no significant heterogeneity of COVID-19 on celiac disease (*Q_Pval* > 0.05). The Egger intercept, approaching zero with *P* > 0.05, indicated no potential horizontal pleiotropy. Furthermore, the MR PRESSO tests demonstrated no horizontal pleiotropy outliers among the genetic IVs, with global test *P* > 0.05 (listed in S3 Table in S2 File).

Our investigation revealed that genetic liability to critical COVID-19 was associated with a higher risk of celiac disease by implementing the IVW approach (odds ratio [OR] = 1.115, 95% confidence interval [CI], 1.007–1.234; *P* = 0.035). This association was consistent with weighted median methods (*P* = 0. 021). However, we did not observe a similar association for COVID-19 infection (IVW: OR = 0.938, 95% CI, 0.438–2.007; *P* = 0.869) or Hospitalized COVID-19 (IVW: OR = 1.082, 95% CI, 0.856–1.367; *P* = 0.510) (Fig 2). Scatter plots (listed in S1 Fig in S1 File) illustrated the effect sizes of associations between COVID-19 and celiac disease as determined by MR methods.

The Steiger test revealed a higher variance ($R^2$) in exposure compared to celiac disease outcome, consolidating the established causal directionality between critical COVID-19 and celiac

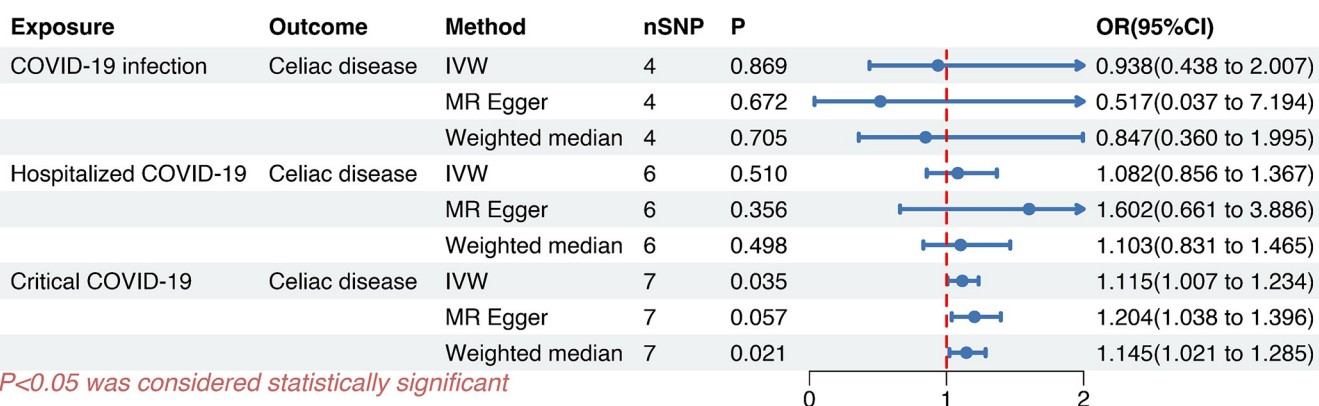

| Exposure | Outcome | Method | nSNP | P | | OR(95%CI) |
|---|---|---|---|---|---|---|
| COVID-19 infection | Celiac disease | IVW | 4 | 0.869 | | 0.938(0.438 to 2.007) |
| | | MR Egger | 4 | 0.672 | | 0.517(0.037 to 7.194) |
| | | Weighted median | 4 | 0.705 | | 0.847(0.360 to 1.995) |
| Hospitalized COVID-19 | Celiac disease | IVW | 6 | 0.510 | | 1.082(0.856 to 1.367) |
| | | MR Egger | 6 | 0.356 | | 1.602(0.661 to 3.886) |
| | | Weighted median | 6 | 0.498 | | 1.103(0.831 to 1.465) |
| Critical COVID-19 | Celiac disease | IVW | 7 | 0.035 | | 1.115(1.007 to 1.234) |
| | | MR Egger | 7 | 0.057 | | 1.204(1.038 to 1.396) |
| | | Weighted median | 7 | 0.021 | | 1.145(1.021 to 1.285) |

*P<0.05 was considered statistically significant*

**Fig 2. Causal effects of COVID-19 infection, hospitalized COVID-19, and critical COVID-19 on celiac disease.** *P* < 0.05 denotes the causal relationship between COVID-19 and celiac disease.

disease (listed in S3 Table in S2 File). Leave-one-out analysis showed the effect of individual SNPs on overall causality, as illustrated in S2 Fig in S1 File. For the assessment of the effect sizes of individual SNPs of COVID-19 on celiac disease, forest plots were employed (listed in S3 Fig in S1 File), providing detailed representations of the impact of individual SNPs on the relationship between COVID-19 and celiac disease.

## Mediation MR analysis connecting critical COVID-19 and celiac disease via *Victivallaceae* abundance

Considering the causal association revealed in the previous MR analysis between critical COVID-19 and celiac disease (IVW: $\beta_1 = 0.109$, 95% CI, 0.007–0.210; OR = 1.115; $P = 0.035$), we selected critical COVID-19 as the exposure and celiac disease as the outcome for this stage of the analysis, aiming to investigate the potential mediation of gut microbiota. Employing MR methods, we explored the causal relationship between critical COVID-19 and the gut microbiota. Our findings indicated that critical COVID-19 was associated with lower relative abundance of the *Victivallaceae* (IVW: $\beta_2 = -0.119$, 95% CI, −0.222 to −0.016; $P = 0.045$), a gut microbiota family belonging to the order *Victivallales*. The association was also supported by the weighted median approach ($\beta = -0.109$, 95% CI, −0.211 to −0.007; $P = 0.036$), as shown in Table 1.

Subsequently, we identified 5 SNPs associated with *Victivallaceae* abundance, serving as genetic IVs for the exposures, with celiac disease selected as the outcome. Our analysis identified a protective effect of an increased *Victivallaceae* abundance against the risk of celiac disease. This protective effect was supported by the IVW ($\beta_3 = -0.249$, 95% CI, −0.452 to −0.046; OR = 0.779; $P = 0.016$) and weighted median methods ($\beta = -0.269$, 95% CI, −0.514 to −0.025; OR = 0.764; $P = 0.031$) (Table 1). Calculating the mediator effect (0.030) as the product of β2 and β3 (β2×β3), it accounted for 27.18% (β2×β3/β1) of the overall effect. In conclusion, our analysis suggests critical COVID-19 were related to a high risk of celiac disease, potentially mediated by reducing *Victivallaceae* abundance.

**Table 1. Mediation MR of critical COVID-19, *Victivallaceae* abundance, and celiac disease.**

| Exposure | Outcome | Method | SNPs | β (95% CI) | OR (95% CI) | *P* |
|---|---|---|---|---|---|---|
| Critical COVID-19 | Celiac disease | IVW | 7 | 0.109 (0.007, 0.210) | 1.115 (1.007, 1.234) | 0.035 |
| | | MR Egger | | 0.186 (0.038, 0.333) | 1.204 (1.038, 1.396) | 0.057 |
| | | Weighted median | | 0.135 (0.021, 0.250) | 1.145 (1.021, 1.285) | 0.021 |
| | | Simple mode | | 0.104 (−0.118, 0.326) | 1.110 (0.889, 1.386) | 0.393 |
| | | Weighted mode | | 0.138 (0.020, 0.256) | 1.148 (1.020, 1.291) | 0.062 |
| Critical COVID-19 | *Victivallaceae* abundance | IVW | 12 | −0.119 (−0.222, −0.016) | 0.888 (0.801, 0.984) | 0.045 |
| | | MR Egger | | −0.132 (−0.261, −0.004) | 0.876 (0.770, 0.996) | 0.071 |
| | | Weighted median | | −0.109 (−0.211, −0.007) | 0.897 (0.810, 0.993) | 0.036 |
| | | Simple mode | | −0.114 (−0.327, 0.100) | 0.892 (0.721, 1.105) | 0.318 |
| | | Weighted mode | | −0.071 (−0.154, 0.012) | 0.931 (0.857, 1.012) | 0.092 |
| *Victivallaceae* abundance | Celiac disease | IVW | 5 | −0.249 (−0.452, −0.046) | 0.779 (0.636, 0.955) | 0.016 |
| | | MR Egger | | 0.076 (−1.249, 1.402) | 1.079 (0.287, 4.063) | 0.917 |
| | | Weighted median | | −0.269 (−0.514, −0.025) | 0.764 (0.598, 0.976) | 0.031 |
| | | Simple mode | | −0.295 (−0.616, 0.026) | 0.745 (0.540, 1.026) | 0.146 |
| | | Weighted mode | | −0.294 (−0.594, 0.006) | 0.745 (0.552, 1.006) | 0.127 |

Note: *P* < 0.05 represents the causal association.

Abbreviations: MR, Mendelian randomization; COVID-19, coronavirus disease 2019; SNP, single-nucleotide polymorphism; β, the regression coefficient; CI, confidence interval; OR, odds ratio; *P*, *P* Value; IVW, inverse variance weighted

## Discussion

In this study, we conducted a two-step MR analysis to explore the causal relationship between COVID-19 and the susceptibility to celiac disease. The comprehensive analysis provided compelling evidence supporting a causal relationship between critical COVID-19 and the risk of celiac disease, consistent with a previous retrospective cohort study [8]. A subsequent mediation MR analysis provided further evidence, revealing that critical COVID-19 heightens the risk of celiac disease by decreasing *Victivallaceae* abundance. Consequently, this research makes a significantly contribution to unraveling the underlying causal mechanisms between COVID-19 and celiac disease.

The observed causal relationship between critical COVID-19 and increased susceptibility to celiac disease illuminates the multifaceted consequences of viral infections on the immune system and autoimmune disorders [30]. Severe acute respiratory syndrome coronavirus 2 (SARS-CoV-2), the virus responsible for COVID-19, has been shown to exert profound effects on lymphocytes [31–34]. Abnormal immune responses play a central role in the celiac disease triggered by gluten ingestion, contributing to inflammation and intestinal damage [35–37]. Regulatory T cells (Tregs) are crucial in maintaining immune tolerance and preventing autoimmunity. A previous study found a reduction in SARS-CoV-2-reactive Tregs in hospitalized COVID-19 patients [34]. In celiac disease, Treg function dysfunction contributes to the loss of tolerance to gluten [38]. The impact of COVID-19 on Tregs and the observed impaired functionality of Tregs in celiac individuals, suggest that COVID-19 may contribute to an environment favoring autoimmune responses.

Celiac disease is characterized by villous atrophy and the presence of autoantibodies targeting transglutaminase 2 (TG2), an enzyme that deamidates gluten. Human Leukocyte Antigen (HLA) genes, particularly HLA-DQ2.5, HLA-DQ 2.2, and HLA-DQ8, are recognized genetic risk factors for celiac disease [39, 40]. In celiac disease patients, antigen-presenting cells express HLA-DQ molecules associated with the disease [41]. Deamidation of gluten peptides by TG2 results in a negative charge, leading to tighter binding with the disease-associated HLA molecule, forming a stable peptide-HLA complex [42–44]. The peptide-HLA complex triggers the activation of gluten-specific CD4+ T cells, resulting in the production of pro-inflammatory cytokines [45, 46]. It also interacts with B cells to induce antibody production, ultimately leading to the destruction of the intestinal villi. Additionally, various non-HLA risk loci, many of which are associated with the functions of T and B cells, have been implicated in celiac disease [24, 47]. Viral infections and HLA gene variants have strongly interaction [48–50]. Critical COVID-19 may exacerbate the genetic predisposition to celiac disease by affecting HLA-DQ molecules.

Considering the latent period of celiac disease, typically characterized by a silent interval lasting several years before symptomatic manifestation, our attention is directed towards exploring the potential impact of external factors, particularly critical COVID-19, in triggering or exacerbating celiac disease symptoms. During viral infection, the immune system generates a response against specific viral epitopes. Some of these viral epitopes may share similarities with self-antigens in the host, known as molecular mimicry [51]. Cross-reactivity occurs when immune cells, initially activated against viral epitopes, recognize and respond to similar epitopes on host tissues, leading to immune system confusion and autoimmunity [52]. Furthermore, viral infections can cause tissue damage and cell death, resulting in the release of self-antigens not initially targeted by the immune response. The exposure of new self-antigens to the immune system can trigger the expansion of the immune response to encompass these additional epitopes, contributing to epitope spreading. Critical COVID-19 may enhance this molecular mimicry and epitope spreading, potentially triggering celiac symptoms in predisposed individuals.

This study employs the MR approach to avoid uncertainties related to causality. HLA genes are a crucial but not exclusive factor in the development of celiac disease. Earlier investigations have demonstrated an association between celiac disease and gut dysbiosis [53, 54]. Nevertheless, discerning whether gut dysbiosis results from celiac disease or contributes to its development poses a significant challenge.

A recent study showed that genetic variations in the gut microbiota may mediate the pathogenesis of celiac disease by increasing the bacterial arginine/polyamine biosynthesis and L-lysine biosynthesis pathways [55]. Our study suggests that the reduction in *Victivallaceae* abundance could potentially act as a mediator in the relationship between critical COVID-19 and celiac disease. *Victivallaceae* is a gut microbiota family, comprised of the genus *Victivallis*, with a significant association with obesity [56] and the clinical response to anti-PD-1 immunotherapy in cancer patients [57]. The decrease in *Victivallaceae* abundance might disrupt the delicate balance of immune regulation, creating an environment conducive to autoimmune responses. This aligns with the broader understanding that the gut microbiota influences on immune function and autoimmune diseases [58, 59].

SARS-CoV-2 can impair the integrity of the intestinal barrier, thereby promoting the translocation of microbial and endotoxin components [60–63]. Elevated levels of TNF-α, CXCL10, CCL2, and IL-10 in COVID-19 patients correlate with reduced populations of specific intestinal bacterial, which suggests that gut dysbiosis may contribute significantly to the development of intense and widespread inflammation [64, 65]. Subsequently, inflammatory factors increase the permeability of the intestinal mucosa by damaging intestinal epithelial cells, leading to impairment of the intestinal barrier. The compromised integrity of the intestinal barrier heightens the probability of immune cells being exposed to gluten peptides. This exposure may initiate or intensify the autoimmune response characteristic of celiac pathogenesis. Furthermore, in individuals susceptible to celiac disease, gliadin can induce the excessive growth of pathogenic intestinal bacteria, leading to gut dysbiosis and forming a vicious cycle [66].

This study has several strengths. Firstly, the COVID-19, *Victivallaceae* abundance and celiac disease data used in our investigation were derived from GWAS, one of the largest genetic studies to date. This choice ensured the robust statistical power for the results and conclusions. Secondly, potential confounding due to population stratification was eliminated by restricting the analyses to populations of European ancestry for both the COVID-19 genetic IVs, *Victivallaceae* abundance genetic IVs, and celiac disease GWAS. Additionally, we successfully identified several genetic variants robustly linked to critical COVID-19 and *Victivallaceae* abundance. Using these variants as instruments, we achieved a strong instrument strength for MR analysis. Furthermore, employing to various analytical methods, there was no discernible pleiotropy or heterogeneity in COVID-19 genetic IVs. Finally, the results of our study present original contributions to the field. The elucidation of *Victivallaceae* abundance in mediating the effect of critical COVID-19 on celiac disease risk holds promise for advancing understanding of the microbiological mechanisms.

Several limitations warrant consideration in interpreting the findings of this study. Firstly, our investigation focused on restricted subset of gut microbiota, revealing a significant association between *Victivallaceae* abundance and celiac disease. This conclusion is drawn from the analysis of gut microbiota data comprising 211 taxa, as presented by Kurilshikov et al. [23]. However, it is essential to notice that gut microbiota include more than just these 211 taxa. Existing literature suggests that a variety of gut microbiota exhibit associations with celiac disease [15, 16]. The possibility of overlooking crucial gut microbiota cannot be discounted. Our findings highlight the significance of the gut microbiota in celiac disease, providing impetus for further exploration though randomized controlled trials and mechanistic investigations to clarify the pathways of the entire spectrum gut microbiota. Secondly, our study discovered

that critical COVID-19 can increase the risk of celiac disease. However, this correlation was not observed in hospitalized COVID-19 and COVID-19 infection. Further researches are imperative to elucidate the causal relationship between hospitalized COVID-19 and COVID-19 infection. Thirdly, the variability in results obtained through different MR methods may be attributed to inherent factors. Notably, MR Egger may be susceptible to factors, such as insufficient statistical efficacy, susceptibility to outlying SNPs and pleiotropy. Finally, it is essential to acknowledge that our study predominantly included individuals of European ancestry. Caution is advised when extrapolating our findings to other ancestral populations, and the external validity of the MR results should be considered in this context.

## Conclusion

This study utilized the available GWAS database to select robust genetic variations as IVs. We elucidated the causal relationship between COVID-19 and celiac disease, while exploring the mediating effect of the gut microbiota, using a two-step Mendelian randomization approach. In line with the findings from a recent observational investigation, our results unravel a compelling association between critical COVID-19 and an elevated risk of celiac disease [8]. Further analysis revealed that critical COVID-19 heightens the risk of celiac disease by decreasing *Victivallaceae* abundance. These discoveries underscore the interactions between critical COVID-19, the gut microbiota and the risk of celiac disease. Future researches should delve deeper into the specific molecular pathways linking viral infections, gut dysbiosis, and autoimmune responses.

## Supporting information

**S1 File.**
(DOCX)

**S2 File.**
(XLSX)

## Acknowledgments

We extend our appreciation to three anonymous reviewers for their invaluable and constructive comments. We would like to thank the COVID-19 Host Genetics Initiative and the MiBio-Gen consortium sharing the GWAS data on COVID-19 and gut microbiota summary statistics. Our gratitude also extends to the IEU open GWAS project (https://gwas.mrcieu.ac.uk/datasets/) for providing available the summary results data of celiac disease.

## Author Contributions

**Conceptualization:** Yuntian Chen, Zhihua Wang, Huaqi Guo.

**Data curation:** Yuxin Zou, Lifeng Yan.

**Formal analysis:** Manyi Pan, Weining Xiong.

**Investigation:** Yuxin Zou, Manyi Pan, Tianyu Zhou, Lifeng Yan, Junjie Yun.

**Methodology:** Yuxin Zou, Yuntian Chen, Junjie Yun, Kai Zhang, Weining Xiong.

**Project administration:** Manyi Pan, Weining Xiong.

**Resources:** Tianyu Zhou, Junjie Yun.

**Software:** Yuxin Zou, Lifeng Yan, Yuntian Chen, Kai Zhang.

**Supervision:** Huaqi Guo, Kai Zhang.

**Validation:** Tianyu Zhou.

**Visualization:** Weining Xiong.

**Writing – original draft:** Yuxin Zou.

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
