## [Decision Letter · Decision Letter 0]

11 Dec 2023

PONE-D-23-36003Critical COVID-19, Victivallaceae Abundance, and Celiac disease: A Mediation Mendelian Randomization StudyPLOS ONE

Dear Dr. Guo,

Thank you for submitting your manuscript to PLOS ONE. After careful consideration, we feel that it has merit but does not fully meet PLOS ONE’s publication criteria as it currently stands. Therefore, we invite you to submit a revised version of the manuscript that addresses the points raised during the review process.

We look forward to receiving your revised manuscript.

Kind regards,

Yasin Sahin

Academic Editor

PLOS ONE

Journal Requirements:

5. Please remove your figures from within your manuscript file, leaving only the individual TIFF/EPS image files, uploaded separately. These will be automatically included in the reviewers’ PDF.

Additional Editor Comments:

Thank you for the study.

I regret to inform you that it cannot be accepted in its current form. However, we can consider a revised version addressing the raised points.

Reviewers' comments:

Reviewer's Responses to Questions

**Comments to the Author**

1. Is the manuscript technically sound, and do the data support the conclusions?

Reviewer #1: No

Reviewer #2: Yes

Reviewer #3: No

2. Has the statistical analysis been performed appropriately and rigorously? 

Reviewer #1: I Don't Know

Reviewer #2: Yes

Reviewer #3: I Don't Know

3. Have the authors made all data underlying the findings in their manuscript fully available?

Reviewer #1: Yes

Reviewer #2: Yes

Reviewer #3: Yes

4. Is the manuscript presented in an intelligible fashion and written in standard English?

Reviewer #1: No

Reviewer #2: Yes

Reviewer #3: No

5. Review Comments to the Author

Reviewer #1: _English editing is highly recommended.

_Why don’t u focus your article on the main issues which are COVID-19 and celiac in general, then u can consider gut dysbiosis and Victivallaceae Abundance as a hypothesis to be as risk factor. Or instead try to propose a theory of gut dysbiosis to promote celiac in critical COVID-19 patients. Hence a significant revision needed in overall title, abstract, significant related data of results and to be thoroughly discussed in discussion with related contrary/supporting previous studies, with concluded message to your readers of the exact relation between COVID-19 and celiac.

Reviewer #2: Dear Editor,

The investigators aimed to investigate a potential ling between COVID infection, gut microbiota and celiac disease.

I would suggest publishing it after addressing some minor points.

Thanks

Queries:

1. No need to keep the sentences from 231 to 236. It is not adding any strength to the discussion or manuscript.

2. Literature is not clear that severity of COVID is closely related to post-COVID syndrome. Even the asymptomatic or mildly symptomatic patients can develop long COVID symptoms. Instead of mentioning post-COVID syndrome, authors might focus more on the critical COVID infection related gut microbiome changes and its potential effect.

3. Could the authors add a small discussion about the role of the presence of genes (particularly HLA) to develop celiac disease? Can we claim critical COVID infection may trigger/ exacerbate the symptoms of celiac disease instead? Given the fact that most celiac disease patients go silent for years until they develop some supporting symptoms, this would wroth discussing!

The authors discussed their strength and limitations very well.

Reviewer #3: The authors worked very hard in preparing this manuscript. Although, It would be very premature to think that only Victivallaceae is associated with Celiac disease. They have also reported unusually high no. of cases as compared to controls, which I think is problematic.

6. PLOS authors have the option to publish the peer review history of their article (what does this mean?). If published, this will include your full peer review and any attached files.

Reviewer #1: No

Reviewer #2: No

Reviewer #3: No

---

## [Author Response · Author response to Decision Letter 0]

23 Jan 2024

More specific information is detailed in the Response to Reviewers.docx

Reply to Review 1

 To address the concerns raised by Reviewer 1 regarding English editing and the focus of the article, we acknowledge the importance of enhancing the clarity and coherence of our manuscript and the focus of gut dysbiosis. 

1. English editing is highly recommended.

 Response: We thank the reviewer for raising the valuable suggestion. Regarding the recommendation for English editing, we want to express our sincere commitment to ensuring the highest standard of language in our manuscript. Recognizing the importance of this aspect, we have sought professional assistance from the company, Editage. In the attached document, you will find certificate of the editing performed on the manuscript. 

2. Focusing the article on the main issues of COVID-19 and celiac disease, and then a hypothesis of gut dysbiosis and Victivallaceae abundance as risk factors

 Response: We thank the reviewer for raising the valuable suggestions. We highly agree with the suggestion to emphasize gut dysbiosis and Victivallaceae Abundance as a potential risk factor in the development of celiac disease in critical COVID-19 patients. This perspective has been incorporated into our revised manuscript. We implemented a major revision, reevaluating the general title, abstract, correlative data presentation, and the deep discussion with previous relevant studies. 

 The abstract and conclusion explicitly establish the connection between COVID-19 and celiac disease for the readers. COVID-19 enhances susceptibility to celiac disease through a reduction in the abundance of Victivallaceae abundance. Detailed information can be found in Bold in Revised Manuscript with Track, specifically line 27 to 32 and 289 to 297.

 In the introduction, we emphasized that genetic factors and a gluten-rich diet are not the only factors in the development of celiac disease. Previous studies have emphasized the link between gut dysbiosis and the development of celiac disease[1, 2]. Understanding the potential influence of COVID-19 on the development or exacerbation of celiac disease, particularly through mechanisms involving gut dysbiosis, has significant implications. Detailed information can be found in Bold in Revised Manuscript with Track, specifically line 47 to 55.

 Our study suggests that the reduction in Victivallaceae abundance could potentially act as a mediator in the relationship between critical COVID-19 and celiac disease. The decrease in Victivallaceae abundance might disrupt the delicate balance of immune regulation, creating an environment conducive to autoimmune responses. We further discuss, the potential mechanisms between SARS-CoV-2, gut dysbiosis and celiac disease. SARS-CoV-2 compromises the integrity of the intestinal barrier[3-6]. Reduced populations of specific intestinal bacteria in COVID-19 patients correlate with elevated levels of inflammatory factors [7, 8]. This suggests that gut dysbiosis may contribute significantly to the development of intense and widespread inflammation. Subsequently, inflammatory factors damage intestinal epithelial cells, leading to impairment of the intestinal barrier. The compromised integrity of the intestinal barrier heightens the probability of immune cells being exposed to gluten peptides. This exposure may initiate the autoimmune response characteristic of celiac pathogenesis. Furthermore, in individuals susceptible to celiac disease, gliadin can induce the excessive growth of pathogenic intestinal bacteria, leading to gut dysbiosis and forming a vicious cycle[9]. Detailed information can be found in Bold in Revised Manuscript with Track, specifically line 243 to 261.

Reply to Review 2

 Regarding the points raised by Reviewer 2, we appreciate the guidance on specific sentences and the emphasis on key gut microbiome changes associated with critical COVID infection.

1. No need to keep the sentences from 231 to 236. It is not adding any strength to the discussion or manuscript.

 Response: We thank the reviewer for raising the valuable suggestion. In the revised version, we have carefully reviewed and omitted those unnecessary sentences. 

2. Literature is not clear that severity of COVID is closely related to no post-COVID syndrome. Even the asymptomatic or mildly symptomatic patients can develop long COVID symptoms. Instead of mentioning post-COVID syndrome, authors might focus more on the critical COVID infection related gut microbiome changes and its potential effect.

 Response: We thank the reviewer for raising the valuable suggestions. Regarding the observation about the clarity in the literature regarding the relationship between COVID-19 severity and post-COVID syndrome, we appreciate the feedback. We have redirected our focus to explore gut microbiome changes with gut microbiome changes and their potential implications, instead of an emphasis on long COVID symptoms. 

 In the introduction, we found that previous studies have emphasized the link between gut microbiome changes altered (gut dysbiosis) and the development of celiac disease. Understanding the potential influence of COVID-19 on the development or exacerbation of celiac disease, particularly through mechanisms involving gut dysbiosis, has significant implications. Detailed information can be found in Bold in Revised Manuscript with Track, specifically line 47 to 55.

 In the Discussion section, we discuss in depth that a reduction in Victivallaceae abundance may disrupt the delicate balance of immune regulation, creating an environment conducive to autoimmune responses. In addition, we further discuss the potential mechanisms between SARS-CoV-2, intestinal dysbiosis, the gastrointestinal barrier, and celiac disease. SARS-CoV-2 compromises the integrity of the intestinal barrier[3-6]. Previous studies have further demonstrated that gut dysbiosis is present in patients with COVID-19 and may contribute significantly to the development of intense and widespread inflammation[7, 8]. Subsequently, inflammatory factors damage the intestinal epithelial cells, leading to a compromised intestinal barrier. With the integrity of the intestinal barrier compromised, immune cells are more likely to be exposed to gluten peptides. This exposure may trigger an autoimmune response characteristic of celiac disease pathogenesis. In addition, in individuals susceptible to celiac disease, gliadin induce overgrowth of intestinal pathogenic bacteria, leading to intestinal dysbiosis and a vicious cycle[9]. Detailed information can be found in Bold in Revised Manuscript with Track, specifically line 243 to 261.

3. add a small discussion about the role of the presence of genes (particular HLA) to develop celiac disease

 Response: We thank the reviewer for raising the valuable suggestions. We have incorporated a discussion on the role of genes, especially HLA, in the pathogenesis of celiac disease. 

 Human leukocyte antigen (HLA) genes, particularly HLA-DQ2.5, HLA-DQ 2.2 and HLA-DQ8, are recognized genetic risk factors for celiac disease[10, 11]. Antigen-presenting cells in celiac disease patients express HLA-DQ molecules associated with the disease[12]. Deamination of glutathione produces a negative charge that binds to the disease-associated HLA molecules to form stable peptide-HLA complexes[13-15]. The peptide-HLA complex triggers the activation of gluten-specific CD4+ T cells, leading to the production of pro-inflammatory cytokines[16]. In addition, activated CD4+ T cells interact with B cells to induce antibody production, ultimately leading to destruction of the intestinal villi. In addition, a variety of non-HLA risk loci, many of which are associated with T- and B-cells function, have been implicated in celiac disease[17, 18]. Viral infections and HLA gene variants have a strong interaction[19-21]. Critical COVID-19 may exacerbate genetic susceptibility to celiac disease by affecting HLA-DQ molecules. Detailed information can be found in Bold in Revised Manuscript with Track, specifically line 215 to 226.

4.Can we claim critical COVID infection may trigger/exacerbate the symptoms of celiac disease instead? Given the fact that most celiac disease patients go silent for years until they develop some supporting symptoms, this would worth discussing!

 Response: We thank the reviewer for raising the thoughtful suggestions. The insightful comments have significantly contributed to enhancing the quality and rigor of our research. In this study, we employed the Mendelian randomization approach to investigate associations with disease at the genetic susceptibility level. Claiming the exacerbation of symptoms necessitates additional empirical validation through dedicated experiments. We have discussed the potential mechanisms that may trigger or exacerbate symptoms of celiac disease in critical COVID infections.

 During viral infection, the immune system generates a response against specific viral epitopes. Some of these viral epitopes may share similarities with self-antigens in the host, known as molecular mimicry[22]. Cross-reactivity occurs when immune cells, initially activated against viral epitopes, recognize and respond to similar epitopes on host tissues, leading to immune system confusion and autoimmunity[23]. Furthermore, viral infections can cause tissue damage and cell death, resulting in the release of self-antigens not initially targeted by the immune response. The exposure of new self-antigens to the immune system can trigger the expansion of the immune response to encompass these additional epitopes, contributing to epitope spreading. Critical COVID-19 may enhance this molecular mimicry and epitope spreading, potentially triggering celiac symptoms in predisposed individuals. Detailed information can be found in Bold in Revised Manuscript with Track, specifically line 227 to 238.

Reply to Review 3

 For the concerns expressed by Reviewer 3 regarding the association with the Victivallaceae family and the reported case numbers, we genuinely acknowledge the merit of the concerns.

1. It be premature to think that only Victivallaceae is associated with Celiac disease.

 Response: We thank the reviewer for raising the thoughtful suggestion. We appreciate the meticulous consideration given to this aspect. Our investigation reveals a significant association between celiac disease and Victivallaceae abundance, based on the gut microbial data (211 taxa) provided by Kurilshikov A et al[24]. Indeed, the gut microbiota comprises not only the mentioned 211 taxa but also various others. As correctly pointed out by the reviewer, attributing celiac disease to a single gut microbiota is premature. Existing literature indicates an association between celiac disease and a diverse range of gut microbiota[1, 2], without excluding the possibility of a connection with others gut microbiota. We took great care to acknowledge this limitation within the manuscript. This perspective is thoroughly discussed in our revised manuscript. Detailed information can be found in Bold in Revised Manuscript with Track, specifically line 272 to 280. The primary contribution of our study lies in establishing, for the first time, a causal relationship between Victivallaceae and celiac disease. This finding serves to reinforce the theory that gut microbiota plays a role in promoting the development of celiac disease. 

2. Unusually high no. of cases as compared to controls.

 Response: We thank the reviewer for raising the thoughtful suggestion. I would like to clarify that the error in the reported number of cases is indeed due to a mistake in our writing process. We recognize the importance of accurate reporting and sincerely apologize for any confusion caused by this oversight. Rest assured, this error has been rectified in the revised manuscript, and we provide the correct and precise information. 

 The COVID-19 Host Genetics Initiative (https://www.covid19hg.org/ , Release 7, European ancestry cohorts) provided GWAS summary statistics for severe respiratory symptom (A2: critically ill COVID vs. population), hospitalization (B2: hospitalized COVID vs. population) and COVID-19 infection (C2: COVID vs. population). The GWAS dataset comprised 13,769 cases vs. 1,072,442 controls (A2), 32,519 cases vs. 2,062,805 controls (B2), and 122,616 cases vs. 2,475,240 controls (C2). Summary statistics for gut microbial taxa were obtained from an extensive GWAS analysis involving 18,340 participants with diverse ancestries and ages across 24 cohorts. The available at the www.mibiogen.org. Summary statistics for Celiac disease were obtained from most comprehensive GWAS to date, analyzing 523,399 SNPs within a cohort comprising 4,533 celiac disease patients and 10,750 healthy individuals. Access to a summary of this dataset is freely available at https://gwas.mrcieu.ac.uk/datasets/ieu-a-276 . Detailed information can be found in Bold in Revised Manuscript with Track, specifically line 88 to 94 and 104 to 105.

 We genuinely appreciate the constructive feedback provided by each reviewer, and we are committed to undertaking the necessary revisions to improve the quality and impact of our manuscript. Your guidance is instrumental in shaping the contribution of our research to the field.

 Thank you once again for your time and consideration. We look forward to submitting the revised manuscript for your further evaluation.

Best regards,

PhD, Huaqi Guo

Department of Respiratory and Critical Care Medicine, Shanghai Key Laboratory of Tissue Engineering, Shanghai Ninth People's Hospital, Shanghai Jiao Tong University School of Medicine 

Email: guohuaqi814102@163.com

1. Zoghi S, Abbasi A, Heravi FS, Somi MH, Nikniaz Z, Moaddab SY, et al. The gut microbiota and celiac disease: Pathophysiology, current perspective and new therapeutic approaches. Crit Rev Food Sci Nutr. 2022:1-21. Epub 20220926. doi: 10.1080/10408398.2022.2121262. PubMed PMID: 36154539.

2. Akobeng AK, Singh P, Kumar M, Al Khodor S. Role of the gut microbiota in the pathogenesis of coeliac disease and potential therapeutic implications. Eur J Nutr. 2020;59(8):3369-90. Epub 20200710. doi: 10.1007/s00394-020-02324-y. PubMed PMID: 32651763; PubMed Central PMCID: PMCPMC7669811.

3. Eleftheriotis G, Tsounis EP, Aggeletopoulou I, Dousdampanis P, Triantos C, Mouzaki A, et al. Alterations in gut immunological barrier in SARS-CoV-2 infection and their prognostic potential. Front Immunol. 2023;14:1129190. Epub 20230315. doi: 10.3389/fimmu.2023.1129190. PubMed PMID: 37006316; PubMed Central PMCID: PMCPMC10050566.

4. Jiao L, Li H, Xu J, Yang M, Ma C, Li J, et al. The Gastrointestinal Tract Is an Alternative Route for SARS-CoV-2 Infection in a Nonhuman Primate Model. Gastroenterology. 2021;160(5):1647-61. Epub 20201209. doi: 10.1053/j.gastro.2020.12.001. PubMed PMID: 33307034; PubMed Central PMCID: PMCPMC7725054.

5. Kariyawasam JC, Jayarajah U, Riza R, Abeysuriya V, Seneviratne SL. Gastrointestinal manifestations in COVID-19. Trans R Soc Trop Med Hyg. 2021;115(12):1362-88. doi: 10.1093/trstmh/trab042. PubMed PMID: 33728439; PubMed Central PMCID: PMCPMC7989191.

6. Cardinale V, Capurso G, Ianiro G, Gasbarrini A, Arcidiacono PG, Alvaro D. Intestinal permeability changes with bacterial translocation as key events modulating systemic host immune response to SARS-CoV-2: A working hypothesis. Dig Liver Dis. 2020;52(12):1383-9. Epub 20200916. doi: 10.1016/j.dld.2020.09.009. PubMed PMID: 33023827; PubMed Central PMCID: PMCPMC7494274.

7. Yeoh YK, Zuo T, Lui GC, Zhang F, Liu Q, Li AY, et al. Gut microbiota composition reflects disease severity and dysfunctional immune responses in patients with COVID-19. Gut. 2021;70(4):698-706. Epub 20210111. doi: 10.1136/gutjnl-2020-323020. PubMed PMID: 33431578; PubMed Central PMCID: PMCPMC7804842.

8. Vabret N, Britton GJ, Gruber C, Hegde S, Kim J, Kuksin M, et al. Immunology of COVID-19: Current State of the Science. Immunity. 2020;52(6):910-41. Epub 20200506. doi: 10.1016/j.immuni.2020.05.002. PubMed PMID: 32505227; PubMed Central PMCID: PMCPMC7200337.

9. Bernardo D, Garrote JA, Nadal I, Leon AJ, Calvo C, Fernandez-Salazar L, et al. Is it true that coeliacs do not digest gliadin? Degradation pattern of gliadin in coeliac disease small intestinal mucosa. Gut. 2009;58(6):886-7. doi: 10

---

## [Decision Letter · Decision Letter 1]

26 Feb 2024

PONE-D-23-36003R1Critical COVID-19, Victivallaceae abundance, and celiac disease: A mediation Mendelian randomization studyPLOS ONE

Dear Dr. Guo,

Thank you for submitting your manuscript to PLOS ONE. After careful consideration, we feel that it has merit but does not fully meet PLOS ONE’s publication criteria as it currently stands. Therefore, we invite you to submit a revised version of the manuscript that addresses the points raised during the review process.

We look forward to receiving your revised manuscript.

Kind regards,

Yasin Sahin

Academic Editor

PLOS ONE

Journal Requirements:

Additional Editor Comments:

Thanks to the authors for the study and appropriate responses to the reviewers. But the following reference about the HLA genes are not associated, please omit that and use the relevant and correct study called ''Frequency of celiac disease and distribution of HLA-DQ2/DQ8 haplotypes among siblings of children with celiac disease. World J Clin Pediatr 2022 July 9; 11(4): 351-359''.

...Human leukocyte antigen (HLA) genes, particularly HLA-DQ2.5, HLA-DQ 2.2 and HLA-DQ8, are recognized genetic risk factors for celiac disease[10, 11]

Reviewers' comments:

Reviewer's Responses to Questions

**Comments to the Author**

1. If the authors have adequately addressed your comments raised in a previous round of review and you feel that this manuscript is now acceptable for publication, you may indicate that here to bypass the “Comments to the Author” section, enter your conflict of interest statement in the “Confidential to Editor” section, and submit your "Accept" recommendation.

Reviewer #1: All comments have been addressed

Reviewer #2: All comments have been addressed

2. Is the manuscript technically sound, and do the data support the conclusions?

Reviewer #1: Yes

Reviewer #2: Yes

3. Has the statistical analysis been performed appropriately and rigorously? 

Reviewer #1: I Don't Know

Reviewer #2: I Don't Know

4. Have the authors made all data underlying the findings in their manuscript fully available?

Reviewer #1: Yes

Reviewer #2: Yes

5. Is the manuscript presented in an intelligible fashion and written in standard English?

Reviewer #1: Yes

Reviewer #2: Yes

6. Review Comments to the Author

Reviewer #1: (No Response)

Reviewer #2: Dear Editor,

I thank you the authors for their hard work to address each of our queries. Beside a minor additional query as

"The paragraph starting in the line of 237 should follow the sentence of 240.", I do not have any further suggestion.

I would suggest publishing it as it is.

Best

7. PLOS authors have the option to publish the peer review history of their article (what does this mean?). If published, this will include your full peer review and any attached files.

Reviewer #1: **Yes: **Nourhan Badwei

Reviewer #2: No

---

## [Author Response · Author response to Decision Letter 1]

24 Mar 2024

Subject: Response to Editors and Reviewers for Manuscript ID [PONE-D-23-36003R1] 

Dear Editors and Reviewers,

 I hope this message finds you well. I wish to extend my heartfelt thanks for the time and effort you have dedicated to reviewing our manuscript, titled "Critical COVID-19, Victivallaceae Abundance, and Celiac disease: A Mediation Mendelian Randomization Study", and for providing constructive feedback. Your thoughtful insights have been valuable, and we deeply value the insights and comments you have shared.

 We have thoroughly reviewed each comment and suggestion, and have taken great care to address each point in the revised manuscript. Below, I have outlined our responses to your feedback, detailing the changes we have made.

Reply to Editors

1.The reference about the HLA genes are not associated, please omit that and use the relevant and correct study.

 Response: We are grateful to the editor for their constructive suggestion. We highly agree with the recommendation to modify our references. Accordingly, we have made a revision：Human leukocyte antigen (HLA) genes, particularly HLA-DQ2.5, HLA-DQ 2.2 and HLA-DQ8, are recognized genetic risk factors for celiac disease[39,40]. Detailed information can be found in Bold in Revised Manuscript with Track, specifically line 217 to 218.

2. Changes in references.

 Response: We sincerely appreciate the time and effort editors have dedicated to reviewing my manuscript. Upon further thoroughly review of our manuscript, we found it necessary to update our references.

(1) We added the study by Samasca et al. (2021) (PMID：34485888), which better reflects the high prevalence of celiac disease in COVID-19. Detailed information can be found in Bold in Revised Manuscript with Track, specifically line 43. 

(2) We are moved reference that was no relevant to the advantages of MR randomization. Detailed information can be found in Bold in Revised Manuscript with Track, specifically line 58 and 62.

(3) In the references we have cited, the papers by Christophersen et al. (2022) (PMID：35119226) and Risnes et al. (2021) (PMID：33654213) are related to the immune system in celiac disease rather than COVID-19. Therefore, we have incorporated these references to support the assertion that "Abnormal immune responses play a central role in the celiac disease triggered by gluten ingestion, contributing to inflammation and intestinal damage", and have eliminated any incorrect citations accordingly. Detailed information can be found in Bold in Revised Manuscript with Track, specifically line 209 to 210.

(4) We added the study by De et al. (2017) (PMID：28913337) to further support the CD4+ T-cells argument. Detailed information can be found in Bold in Revised Manuscript with Track, specifically line 222.

(5) We have rewritten the conclusion section to more clearly articulate the methods and findings of this study, along with the significance of the research, and have removed irrelevant references. Detailed information can be found in Bold in Revised Manuscript with Track, specifically line 286 to 293.

3. Modifications in some part of the manuscript:

Response: We sincerely appreciate the time and effort editors have dedicated to reviewing my manuscript. After a thorough review of our manuscript, we have implemented several modifications aimed at enhancing the clarity, accuracy, and significance of our study.

(1) In accordance with the journal's guidelines, the *corresponding authors should be listed in a specific order: starting with the primary corresponding author, followed by the second, and then the third. Consequently, we have realigned the order of the corresponding authors in the *Corresponding Author to reflect the order indicated by the title's listing of corresponding authors. We would like to clarify that this realignment has been made solely within the *Corresponding Author. The order of the authors, including the corresponding authors, remains consistent with our original submission. Detailed information can be found in Bold in Revised Manuscript with Track, specifically line 12 to 14.

(2) Based on Mendelian randomization literature[1], we more accurately described "the independence assumption" as: genetic variants should be independent of any confounders that may affect the relationship between the exposure and outcome. Detailed information can be found in Bold in Revised Manuscript with Track, specifically line 67 to 68. Additionally, "the exclusion restriction assumption" should emphasize that SNPs affect the outcome only through the exposure pathway. Hence, we revised "SNPs solely influence the outcome through the exposure pathway" to "SNPs influence the outcome solely through the exposure pathway" for clearer articulation of this assumption's meaning. Detailed information can be found in Bold in Revised Manuscript with Track, specifically line 69. 

(3) The F-statistics should be in the range of 53.07255 to 183.849(listed in S2 Table in S2 File). Accordingly, line 153 in the manuscript should be: "F-statistics ranging from 53.07 to 183.85". Detailed information can be found in Bold in Revised Manuscript with Track, specifically line 153. 

(4) Exposures and outcomes are considered causally related at a significance level of P < 0.05. It is the IVW (P = 0.035) and weighted median (P = 0.021) methods that could conclude that critical COVID-19 is associated with a higher risk of celiac disease, as demonstrated in Figure 2. Detailed information can be found in Bold in Revised Manuscript with Track, specifically line 158 to 160. 

(5) The results of the Leave-one-out analysis demonstrated the causal relationships of individual SNPs in COVID-19 infection, COVID-19 hospitalize, and critical COVID-19 with celiac disease (S2 Fig in S1 File). According to the conclusions drawn from the article, there is no causal relationship between COVID-19 infection or hospitalized COVID-19 and celiac disease. Therefore, "Leave-one-out analysis showed the effect of individual SNPs on overall causality" could more accurately describes the results of the Leave-one-out analysis. Detailed information can be found in Bold in Revised Manuscript with Track, specifically line 171.

(6) Upon reviewing our submitted manuscript and reconfirming all the included analysis and data, we realized that there is a minor typographical error in Table 1 of the manuscript. Specifically, the error lies in the P value using the weighted mode method when analyzing the causal effect between Victivallaceae abundance and celiac disease. This was an inadvertent typographical mistake and we apologize for any confusion it may have caused. I would like to assure you that this error does not alter the findings or conclusions drawn from our study. All analyses and results presented are based on the correct data, and the integrity of our research remains intact. We have rectified at Table 1. Detailed information can be found in Bold in Revised Manuscript with Track, specifically Table 1 and line 185.

(7) We have rewritten the conclusion section to more clearly elucidate the methods and results of this study, while also emphasizing the significant implication of the research. Detailed information can be found in Bold in Revised Manuscript with Track, specifically line 286 to 293.

Reply to Review 1

Response: We would like to express my deepest gratitude for the valuable time and effort you have dedicated to reviewing our manuscript.

Reply to Review 2

1. The paragraph starting in the line of 237 should follow the sentence of 240.

Response: We would like to express my deepest gratitude for the valuable time and effort you have dedicated to reviewing our manuscript. Your constructive suggestion has been valuable to enhancing the quality of our work.

 In the revised version, we have positioned the paragraph starting from line 237 to follow the sentence on line 240 in the manuscript. Your expertise has significantly contributed to the improvement of our manuscript.

This study employs the MR approach to avoid uncertainties related to causality. HLA genes are a crucial but not exclusive factor in the development of celiac disease. Earlier investigations have demonstrated an association between celiac disease and gut dysbiosis. Nevertheless, discerning whether gut dysbiosis results from celiac disease or contributes to its development poses a significant challenge. 

Detailed information can be found in Bold in Revised Manuscript with Track, specifically line 238 to 241.

 We genuinely appreciate the constructive feedback provided by editors and reviewers. We have improved our manuscript according to your valuable suggestions. These modifications have significantly enhanced the quality and clarity of our work.

 Thank you once again for your time and consideration. We look forward to your further evaluation and hope that our revised manuscript meets the esteemed standards of PLOS ONE.

Best regards,

PhD, Huaqi Guo

Department of Respiratory and Critical Care Medicine, Shanghai Key Laboratory of Tissue Engineering, Shanghai Ninth People's Hospital, Shanghai Jiao Tong University School of Medicine 

Email: guohuaqi814102@163.com

1. Sekula P, Del Greco MF, Pattaro C, Kottgen A. Mendelian Randomization as an Approach to Assess Causality Using Observational Data. J Am Soc Nephrol. 2016;27(11):3253-65. Epub 20160802. doi: 10.1681/ASN.2016010098. PubMed PMID: 27486138; PubMed Central PMCID: PMCPMC5084898.

---

## [Editor Report · Decision Letter 2]

27 Mar 2024

Critical COVID-19, Victivallaceae abundance, and celiac disease: A mediation Mendelian randomization study

PONE-D-23-36003R2

Dear Dr. Huaqi Guo, 

Thank you for the study. I think that it will contribute to the literature.

We’re pleased to inform you that your manuscript has been judged scientifically suitable for publication and will be formally accepted for publication once it meets all outstanding technical requirements.

Kind regards,

Yasin Sahin

Academic Editor

PLOS ONE

Additional Editor Comments (optional):

Thank you for the study. I think that it will contribute to the literature